# Diversity of Anaplasmataceae Transmitted by Ticks (Ixodidae) and the First Molecular Evidence of *Anaplasma phagocytophilum* and *Candidatus* Anaplasma boleense in Paraguay

**DOI:** 10.3390/microorganisms12091893

**Published:** 2024-09-14

**Authors:** Oscar Daniel Salvioni Recalde, Miriam Soledad Rolón, Myriam Celeste Velázquez, Martin M. Kowalewski, Jorge Javier Alfonso Ruiz Diaz, Antonieta Rojas de Arias, Milton Ozório Moraes, Harrison Magdinier Gomes, Bruna de Azevedo Baêta, Matheus Dias Cordeiro, María Celeste Vega Gómez

**Affiliations:** 1Center for the Development of Scientific Research (CEDIC), Manduvira 635, Asunción 1255, Paraguay; danioni87@gmail.com (O.D.S.R.); rolonmiriam@gmail.com (M.S.R.); jorwish@gmail.com (J.J.A.R.D.); rojasdearias@gmail.com (A.R.d.A.); 2Programa de Pós-Graduação em Biologia Celular e Molecular, Instituto Oswaldo Cruz (IOC), Fundação Oswaldo Cruz (FIOCRUZ), Rio de Janeiro 21040-900, Brazil; 3Investigación y Conservación, Fundación Moisés Bertoni, Arguello 208, Asunción 1255, Paraguay; mvelazquez@mbertoni.org.py; 4Estación Biológica Corrientes CECOAL (Centro de Ecología Aplicada del Litoral) CONICET-UNNE, Corrientes 3400, Argentina; martinkow@gmail.com; 5Laboratório de Biologia Molecular Aplicada à Micobactérias, Instituto Oswaldo Cruz (IOC), Fundação Oswaldo Cruz (FIOCRUZ), Av. Brasil 4365, Manguinhos 21045-900, Brazil; magdinier@gmail.com; 6Laboratory of Parasitic Diseases of the Federal Rural University of Rio de Janeiro (UFRRJ), Km 07, Seropédica, Rio de Janeiro 21040-900, Brazil; babaeta@hotmail.com (B.d.A.B.); mathcordeiro@hotmail.com (M.D.C.)

**Keywords:** *Anaplasma* spp., *Ehrlichia* spp., *Wolbachia* spp., tick-borne diseases, free-living ticks, high-resolution melting

## Abstract

Anaplasmataceae bacteria are emerging infectious agents transmitted by ticks. The aim of this study was to identify the molecular diversity of this bacterial family in ticks and hosts, both domestic and wild, as well as blood meal sources of free-living ticks in northeastern Paraguay. The bacteria were identified using PCR-HRM, a method optimized for this purpose, while the identification of ticks and their blood meal was performed using conventional PCR. All amplified products were subsequently sequenced. The bacteria detected in the blood hosts included *Ehrlichia canis*, *Anaplasma platys*, and *Anaplasma phagocytophilum*, *Candidatus* Anaplasma boleense, and *Wolbachia* spp., which had not been previously reported in the country. Free-living and parasitic ticks on dogs (*Canis lupus familiaris*) and wild armadillos (*Dasypus novemcinctus*) were collected and identified as *Rhipicephalus sanguineus* and *Amblyomma* spp. The species *E. canis*, *A. platys*, *A. phagocytophilum*, and *Ca*. A. boleense were detected in domestic dog ticks, and *E. canis* and *A. platys* were found for the first time in armadillos and free-living ticks. Blood feeding sources detected in free-living ticks were rodents, humans, armadillos and dogs. Results show a high diversity of tick-borne pathogens circulating among domestic and wild animals in the northeastern region of Paraguay.

## 1. Introduction

Vector-borne diseases (VBDs) are a group of diseases caused by various species of bacteria, parasites, and viruses [1]. These pathogens can be transmitted by arthropod vectors, such as mosquitoes, sand flies, triatomine bugs, and ticks, and other groups of animals, such as mollusks [2]. According to the World Health Organization, VBDs are responsible for about 17% of all infectious diseases and can cause more than 700,000 deaths each year [1]. In addition to a negative impact on public health, the global increase in the incidence of VBDs in recent years has caused considerable economic damage to affected countries, and thus, designing strategies to control this group of diseases is a major challenge in sustainable health goals [3,4].

Regarding ticks, in addition to the epidermal damage that toxins in their saliva can cause, they are considered one of the most relevant arthropods to public health, surpassed only by mosquitoes in their role as human pathogen vectors due to their ability to transmit infectious agents [5]. Ticks are hematophagous ectoparasites of terrestrial vertebrates that are widely spread throughout the world. They are classified as arthropods of the class Arachnida, divided into the following two main families: Argasidae (soft ticks) and Ixodidae (hard ticks). The world’s largest number of species reported belong to the last family, including the most medically important species [5,6]. The emergence and re-emergence of tick-borne diseases (TBDs) has been facilitated by several factors related to human behavior, such as the continued exploitation of environmental resources, the invasion of wildlife habitats, and an increase in outdoor activities [6,7].

The vast repertoire of pathogens transmitted by ticks includes bacteria. In this context, it is essential to emphasize that these arthropods play a fundamental role in the evolution and transmission of the Anaplasmataceae family bacteria, which are causative agents of emerging infectious diseases [8]. The Anaplasmataceae family includes Gram-negative obligate intracellular bacteria that can infect vertebrate animals, causing diseases such as anaplasmosis and ehrlichiosis in humans and animals [9,10,11]. The *Ehrlichia* genus comprises several species, including *E. canis*, *E. chaffeensis*, *E. ewingii*, *E. muris*, *E. ruminantium*, and *E. minasensis* [12]. The *Anaplasma* genus comprises *A. marginale*, *A. centrale*, *A. ovis*, *A. bovis*, *A. phagocytophilum*, *A. platys*, and other species that have not yet been definitively classified [13].

Numerous researchers globally have documented eco-epidemiological, biological, and molecular features of the Anaplasmataceae family present in ticks due to their potential threat to human and animal health [14,15,16,17,18]. It is essential to study this family’s genetic diversity, prevalence, phylogenetic characteristics, and presence in both the vector and the host in order to obtain a better understanding of their epidemiologic significance [11,19].

Studies have been conducted in Latin America to characterize bacterial species transmitted by ticks using molecular techniques such as PCR, serological methods, and eco-epidemiological studies [10,20,21,22]. There is evidence that veterinarians are at a high risk of exposure to pathogens of the Anaplasmataceae family due to the frequency of infection in dogs and other canids [10]. In addition, a significant bacterial infection rate has been detected in livestock, resulting in significant financial losses in the agricultural industry [21].

In Paraguay, the current understanding of TBDs is limited in terms of the range of pathogens that are present in these arachnids and the role that they play as vectors [23]. In Asunción, the presence of *E. canis* in domestic dogs and veterinary personnel with occupational exposure was reported by Tintel et al. [24]. On the other hand, Pérez-Macchi et al. [25] reported the prevalence of *A. platys* and *E. canis* among domestic dogs from urban areas of Asunción and recently documented a case of a dog infected with *A. platys* [26]. These findings highlight the importance of conducting studies in the country using this approach to improve knowledge of zoonotic diseases that affect public health.

Based on the gap in knowledge regarding TBDs, we conducted this research in the Mbaracayú Forest Biosphere Reserve (MFBR). This protected area represents two ecosystems inhabited by indigenous Aché and Guaraní communities [27]. The diversity of hosts, coexistence with human populations, and environmental conditions facilitate the spread of ticks and enhance the transmission potential of various pathogens.

This study aimed to investigate the occurrence and diversity of Anaplasmataceae bacteria in ticks, domestic and wild hosts, and tick diversity in different habitats and identify blood meal sources of free-living ticks using molecular techniques.

## 2. Materials and Methods

### 2.1. Study Areas and Sample Collection

Ticks, domestic dogs (*Canis lupus familiaris*), and wild armadillos (*Dasypus novemcinctus*) were sampled between 2018 and 2022 in rural and forest areas of the Mbaracayú Forest Biosphere Reserve (MFBR) and surrounding communities. The MFBR is situated in the Canindeyú department, in the northeastern region of Paraguay, approximately 330 km from the capital, Asunción (Figure 1A). Blood samples were collected by a veterinarian using sterile 10 mL syringes and placed in EDTA tubes (BD Vacutainer**^®^**, Franklin Lakes, NJ, USA) (Figure 1B,C). Furthermore, ticks associated with these animals were collected using entomological tweezers and placed individually in vials containing 70% ethanol. Free-living ticks were collected using the cloth dragging technique in forest fragments, which was based on passing a 1 m^2^ white flannel over the vegetation and checking the flannel for the presence of caught ticks every 5–10 m (Figure 1D), as previously described [28]. The ticks were removed and placed in 1.5 mL tubes containing ethanol.

### 2.2. Classical and Molecular Identification of Tick Species

Adult ticks were identified using a dichotomous key [29,30], whereas the nymphs and larvae of the *Rhipicephalus* and *Amblyomma* genera, were identified based on specific keys [31,32]. To confirm the identification of specimens at the molecular level, a partial region of the *16S rRNA* gene was amplified using the primers and conditions specified by Mangold et al. [33] (Table 1). We selected and amplified at least two specimens for each of the identified genera’s life stages (adult and nymph).

### 2.3. DNA Extraction from Ticks and Host Blood

Genomic DNA was extracted and purified from the blood of dogs and armadillos, as well as ticks, using the commercially available GeneJET Genomic DNA Purification Kit (#K0722-Thermo Scientific**^®^**, Waltham, MA, USA) according to the manufacturer’s instructions. The purity and concentration of the extracted material were evaluated using a DS-11FX DeNovix**^®^** Spectrophotometer. The material was then stored at −20 °C until further use. Before extracting DNA from the tick, its surface was sterilized by treating it with 1% sodium hypochlorite for 5 min. Then, it was washed with 2% chlorhexidine for another 5 min and rinsed three times with sterile water [34]. All ticks collected from hosts were processed individually. Free-living ticks were processed differently, as follows: the adults individually, nymphs in pools of 20, and larvae in pools of about 100 individuals.

**Table 1 microorganisms-12-01893-t001:** List of primers utilized in this study.

Target Gene	Primers Code	Primers (5′→3′)	Amplicon Size	Reference
Ticks DNA *16S rRNA* (PCR)	16S +1	AACGAACGCTGGCGGCAAGC	460 bp	[33]
	16S −1	AGTAYCGRACCAGATAGCCGC		
Vertebrate *cytb* (PCR)	Cyb1	GAAGATGCWGTWGGWTGTACKGC	358 bp	[35]
	Cyb2	AGMGCTTCWCCTTCWACRTCYTC		
Anaplasmataceae *16S rRNA* (PCR-HRM)	EHR16SD	GGTACCYACAGAAGAAGTCC	345 bp	[36]
	EHR16RD	TAGCACTCATCGTTTACAGC		
Anaplasmataceae *groEL* (Nested-PCR)	gro607 F ^a^	GAAGATGCWGTWGGWTGTACKGC	664 bp	[37]
	gro1294 R ^a^	AGMGCTTCWCCTTCWACRTCYTC		
	gro677 F ^b^	ATTACTCAGAGTGCTTCTCARTG	315 bp	
	gro1121 R ^b^	TGCATACCRTCAGTYTTTTCAAC		

^a^ First round of reaction, ^b^ Second round of reaction.

### 2.4. Blood Feeding Source in Free-Living Ticks

To determine the blood meal sources of free-living ticks, 64 available DNA samples (32 females, 25 nymph pools, and 7 larval pools) were analyzed. For this, conventional PCR was used to amplify a 358 bp partial region of the vertebrate-specific cytochrome b (*cytb*) gene described by Boakye et al. [35] (Table 1). PCR reactions were performed using a thermal cycler (Veriti-Applied Biosystems**^®^**) in a final volume of 50 µL containing 25 µL of mix 2× GoTaq Green Master Mix (Promega**^®^**, Madison, WI, USA), 5 µL template DNA at 8–10 ng/µL, and 0.5 µM final concentration of each primer.

### 2.5. Optimization of PCR-HRM for Detecting and Differentiating Genera in the Family Anaplasmataceae

A real-time PCR with a subsequent high-resolution melting (HRM) analysis has been adapted for the simultaneous amplification and differentiation of bacteria of the genera *Anaplasma* spp. and *Ehrlichia* spp. in a single reaction. For this purpose, a partial region of the 345 bp *16S rRNA* gene was amplified using the primers EHR16SD and EHR16RD, as previously described by Parola et al. [36] (Table 1). Each PCR reaction was performed in a final volume of 20 µL containing a 10 µL mix 2× HRM PCR Master Mix (QIAGEN**^®^,** Hilden, Germany), 2 µL of template DNA at 20 ng/µL, and a 0.5 µM final concentration of each primer. The cycling conditions used were initial denaturation at 95 °C for 5 min, followed by 35 cycles of denaturation at 95 °C for 10 s, annealing at 55 °C for 30 s, and extension at 72 °C for 20 s. A Rotor-Gene 6000 thermal cycler (QIAGEN**^®^**, Hilden, Germany) was used. After real-time PCR, amplicon dissociation was immediately started by a melting step in the same machine. The range was set from 81 °C to 88 °C, with a slope of 0.1 uC/s, and 2 s at each temperature. The high-resolution melting curve analysis was performed with the derivative of the raw data after smoothing with the software, version 2.1.0. (QIAGEN**^®^**, Hilden, Germany). The melting temperature (T_m_) of each genus was determined, and the repeatability of the assay was evaluated by performing five independent experiments, each with 3 replicates, with *A. platys* and *E. canis* DNA as positive controls for all reactions and *Rickettsia parkeri* DNA and molecular grade H_2_O as negative controls. The melting values were then averaged, and the standard deviation was calculated. The T_m_ of the different species that were discovered in this study were evaluated using the same methodology.

### 2.6. Molecular Identification of Anaplasmataceae Based on the 16S rRNA and groEL Gene

For the identification and phylogenetic analysis of Anaplasmataceae bacteria, the resulting products from a real-time PCR assay with a high-resolution melting curve analysis (HRM) underwent sequencing. In addition, in order to better characterize bacteria detected in this study, we amplified a partial region of the 315 bp heat shock protein-60 (*groEL*) gene using a nested PCR approach, with primers and reaction conditions previously detailed by Tabara et al. [37] A PCR reaction was performed using a thermal cycler (Veriti-Applied Biosystems**^®^**, Austin, TX, USA) at a final volume of 25 µL. The reaction mixture included 12.5 µL of GoTaq Green Master Mix (2×, Promega**^®^**, Madison, WI, USA), 3 µL of template DNA at 20 ng/µL, and a 0.5 µM final concentration of each primer.

### 2.7. Purification, Sequencing of Amplified Products, and Phylogenetic Analysis

The resulting PCR amplicons were purified with a GeneJET Genomic PCR Purification Kit (#K0702, Thermo Scientific**^®^**, Waltham, MA, USA) according to the manufacturer’s instructions and sequenced (Sanger sequencing carried out by Macrogen**^®^** Company, Seoul, Republic of Korea) to determine the species of Anaplasmataceae bacteria, ticks, and the blood meal sources of free-living ticks. From the results obtained, a consensus sequence was generated using BioEdit software v. 7.0 [38], followed by a BLAST analysis, with sequences deposited in GenBank.

For phylogenetic analysis, the following partial *16S rRNA* and *groEL* gene sequences deposited in GenBank were downloaded: *E. canis 16S rRNA* (NR.118741.1, MG967466.1, EF195135.1) and *groEL* (CP000107.1, OP066708); *E. chaffensis 16S rRNA* (NR074500.2) and *groEL* (CP000236.1); *A. platys 16S rRNA* (KF826284.1, JX893521.1, MH129061.1) and *groEL* (CP046391.1); *Anaplasma phagocytophilum 16S rRNA* (U02521.1, GQ412337. 1, MK814404) and *groEL* (CP035303.1, GQ452227.2); *A. marginale 16S rRNA* (AJ633048.1) and *groEL* (CP001079.1); *A. ovis 16S rRNA* (AJ633049.1) and *groEL* (CP015994.2); *Candidatus* Anaplasma boleense *16S rRNA* (KX987334.1) and *groEL* (OQ509028.1); and *Wolbachia* spp. *16S rRNA* (KP114101.1). Additionally, the *Anaplasma* spp., *16S rRNA* sequence (JQ685509) was used. Sequences from the species *R. rickettsii* for the *16S rRNA* gene (U11021.1) and *Wolbachia pipientis* for the *groEL* gene (CP092140.1) were used as an outgroup.

A phylogenetic tree was constructed using the maximum likelihood method with the Molecular Evolutionary Genetics Analysis MEGA v.7.0 software [39], with 1000 replicates. The K2+G model was used for the *16S rRNA* gene phylogeny, and the T92+G+1 model was used for the *groEL* gene. Sequences generated in this study were submitted to GenBank and assigned accession numbers OR976129 to OR976137 for *16S rRNA* (ticks); OR976117 to OR976128 and PP930511 to PP930512 for *16S rRNA* (Anaplasmataceae); and PP928796 to PP928805 for *groEL* (Anaplasmataceae).

### 2.8. Data Analysis and Mapping

The study area map was generated with ArcGIS v.10.8.2 software. PCR-HRM efficiency was assessed using LinRegPCR v.2021.2 software [40], and the resulting values were exported to an Excel 2016 software (Microsoft**^®^**, Corporation, WA, USA) document for further analysis. The normal distribution of the data was evaluated using the Shapiro–Wilk test. Descriptive statistics were used to calculate presence of pathogens and blood meal sources. Fisher’s exact test with 95% confidence for nonparametric data was used to determine pathogen prevalence. The prevalence of Anaplasmataceae bacteria in ticks analyzed in pools was assessed using the minimum infection rate (MIR) using the formula previously described [41]. Additionally, the prevalence in proportion was estimated using the EpiTools epidemiological calculator this web-based tool is available at (https://epitools.ausvet.com.au/, accessed on 5 December 2023) [42]. All data were analyzed using GraphPad 8 Prism 8.0.1. The sequencing products were analyzed using NCBI BLAST online software (https://blast.ncbi.nlm.nih.gov/Blast.cgi, accessed on 5 December 2023) and BioEdit v. 7.0 software [38].

### 2.9. Ethical Approval

The collection of ticks and blood from domestic and wild hosts was carried out under the authorization of the Ministry of the Environment and Sustainable Development (MADES-Paraguay in Spanish) under scientific collection permits N° 270 and N° 036, respectively. In addition, we obtained written consent from owners to collect blood samples from their domestic dogs.

## 3. Results

### 3.1. Taxonomic Identification of Ticks Collected from Hosts and Free-Living Ticks

A total of 1924 ticks (79 females, 30 males, 768 nymphs, and 1047 larvae) were collected and identified, 1855 of which were free-living ticks and 69 of which were host-collected (Table 2). Of 203 examined dogs, 23 (11.3%) were tick-infested; 26 specimens were collected, 24 were identified as *Rhipicephalus sanguineus* (10 females, 6 males, and 8 nymphs), and 2 females were identified as *Amblyomma ovale* (Table 2). Of a total of 35 wild armadillos, 13 (37.1%) were tick-infested; 43 specimens were collected, 39 were identified as *Amblyomma sculptum* (21 females and 18 nymphs), and 4 nymphs were identified as *Amblyomma coelebs* (Table 2). Of the 1855 free-living ticks, 808 were identified as *A. sculptum* (46 females, 24 males, and 738 nymphs), and 1047 larvae were identified as *Amblyomma* spp. (Table 2).

### 3.2. Blood Feeding Sources of Free-Living Ticks

To determine the food source of free-living ticks, 64 of 116 samples (55.2%) were analyzed, of which 57 were *A. sculptum* (32 females and 25 pooled samples of nymphs) and 7 were *Amblyomma* spp. (larvae). Of these, 22 samples (34.4%) were amplified for the cytochrome b gene (cyt b), confirming the presence of vertebrate blood (Table 3). The post-sequencing analysis showed the presence of DNA from four different food meal sources, the highest percentage being the order Rodentia, with 45.5% (10/22) detected in females of *A. sculptum*, followed by Homo sapiens, with 22.7% (5/22) detected in nymphs of *A. sculptum* (Table 3). The third most frequent blood meal source was *D. novemcinctus*, with 18.2% (4/22), which was detected in one female and in three nymphs of *A. sculptum*. The fourth most frequent blood meal source detected was *Canis lupus familiaris*, with 13.6% (3/22), which was detected in one female and two nymphs of *A. sculptum* (Table 3). All larvae tested were negative.

### 3.3. Optimization of PCR-HRM for the Detection and Differentiation of the Genus of the Anaplasmataceae Family

The PCR-HRM technique using the *16S rRNA* marker showed specificity and reproducibility for the Anaplasmataceae family, with constant melting values in each reaction. After evaluating five independent experiments, the average T_m_ values were 85.28 ± 0.03 °C for *A. platys* and 83.68 ± 0.07 °C for *E. canis* (Figure 2).

### 3.4. Detection and Differentiation of the Anaplasmataceae Family Using HRM in Host Blood and Ticks

The optimized PCR-HRM technique was used to detect and identify bacteria belonging to the Anaplasmataceae family. Amplified fragments were sequenced, and a similarity search in the GenBank database was used to confirm our results and determine the species of bacteria present in the various habitats examined.

In this study, 203 blood samples were collected from domestic dogs and 35 were collected from wild armadillos. In dogs, the prevalence of Anaplasmataceae infection was 22.2% (45/203) [95% CI: 16.7–28.5%], of which 22 samples were identified as *E. canis* (10.8%), 2 were identified as A. platys (1.0%), 6 were identified as *A. phagocytophilum* (3.0%), 3 were identified as *Ca*. A. boleense (1.5%), and 12 were identified as *Wolbachia* spp. (5.9%) (Figure 3). The infection rate among armadillos was 37.1% (13/35) [95% CI: 21.5–55.1%], of which nine samples were infected with *E. canis* (25.7%) and four were infected with *A. platys* (11.4%) (Figure 3).

In ticks, a total of 139 specimens were analyzed individually, and the larval and nymphal stages of free-living ticks were analyzed in 9 and 35 pools, respectively, resulting in a total of 183 samples. The prevalence of Anaplasmataceae in ticks collected from domestic dogs was 34.6% (9/26) [95% CI: 17.2–55.7%]. The only positive species was *R. sanguineus*, with four males infected with *E. canis*, two females infected with *A. platys*, two females infected with *A. phagocytophilum*, and one female infected with *Ca*. A. boleense (Table 4). The prevalence of Anaplasmataceae in ticks collected from wild armadillos was 23.3% (10/43) [95% CI: 11.8–38.6%]. These specimens, identified as *A. sculptum* (six females and four nymphs), were found to be infected with *E. canis* (Table 4).

The prevalence of Anaplasmataceae in the free-living ticks that were individually analyzed was 21.4% (15/70) [95% CI: 12.5–32.9%]. Within this group, *A. sculptum* ticks were found infected with *E. canis* (five females) and *Anaplasma* spp. (eight females and two males) (Table 4). In the immature free-living specimens collected, a prevalence of Anaplasmataceae of 1.29% [95% CI: 0.75–2.04%] was observed in variable groups of the pools analyzed. Exclusively ticks in nymphal stages infected with *E. canis* and *Anaplasma* spp., with respective MIR values of 1.25% and 1.0%, were observed (Table 5). In contrast, larval samples were negative for the presence of bacteria of the Anaplasmataceae family (Table 5).

In addition, no co-infections of the studied bacteria were detected in the positive samples. After species identification by sequencing, a detailed analysis of the HRM results was performed. Five normalized curves with different T_m_ values for each species were obtained. The average values of the T_m_ for each identified species were obtained from three independent repetitions and are detailed as follows: *E. canis*: 83.68 °C; *A. platys*: 85.28 °C; *A. phagocytophilum*: 85.45 °C; *Ca*. A. boleense: 85.08 °C, and *Wolbachia* spp.: 83.78 °C (Figure 4).

### 3.5. Sequence Analysis and Phylogeny of Anaplasmataceae Family

An analysis of the sequenced samples confirmed the circulation in the study area of five species of bacteria from the Anaplasmataceae family, corresponding to *Ehrlichia canis*, *Anaplasma platys*, *Anaplasma phagocytophilum*, *Candidatus* Anaplasma boleense, and *Wolbachia* spp. (Table 6).

Sequences obtained from the bacteria detected in samples CF393 (dog), DN387 (armadillo), and DN387G (*A. sculptum*) were closely related to the sequence corresponding to *E. canis* (MN922610.1 for *16S rRNA* and OP006713.1 for *groEL*), with a similarity between 99.13 and 100% (Table 6). Samples CF391 (dog) and DN389 (armadillo) were closely related to the sequence corresponding to *A. platys* (MN630836.1 for *16S rRNA* and CP046391.1 for *groEL*), with a similarity between 99.10 and 99.43% (Table 6). Samples 9N1 and 3N1 (*A. sculptum*) were closely related to the sequence corresponding to *Anaplasma* spp. (MT019534.1 for *16S rRNA* and LC381241.1 for *groEL*), with a similarity between 85.01 and 100.0% (Table 6). Samples C20P2 (dog) and CF340G (*R. sanguineus*) were closely related to the sequence corresponding to *A. phagocytophilum* (CP015376.1 for *16S rRNA* and OQ453812.1 for *groEL*), with a similarity between 99.42 and 100.0% (Table 6). Sample CF519 (dog) showed a close relationship with the sequence corresponding to *Ca*. A. boleense (KX987335.1 for *16S rRNA* and OQ509028.1 for *groEL*), with a similarity between 99.14 and 99.71% (Table 6). Finally, sample CF472 (dog) showed a close relationship with the sequence corresponding to *Wolbachia* spp. (MT792375.1 for *16S rRNA*), with 99.13% similarity (Table 6). Amplification of the *groEL* gene was unsuccessful for this sample.

The maximum likelihood (ML) tree, which was constructed based on reference sequences and sequences generated in this study, revealed the presence of three well-defined groups within the phylogeny of the *16S rRNA* gene, comprising *Anaplasma* spp., *Ehrlichia* spp., and *Wolbachia* spp., (Figure 5a). In contrast, the phylogeny of the *groEL* gene exhibited the presence of two well-defined groups, with *Anaplasma* spp. and *Ehrlichia* spp. forming a separate cluster (Figure 5b). Phylogeny based on the *16S rRNA* gene showed that the sequences of samples DN389, CF391, 9N1, and 3N1 clustered together with reference sequences of *A. platys* (KF826284.1, MH129061.1, and JX893521.1) (Figure 5a). Regarding the *groEL* gene phylogeny, samples DN389 and CF391 clustered with the *A. platys* reference sequence (CP046391) (Figure 5b), while samples 9N1 and 3N1 formed an independent group (Figure 5b). Furthermore, samples C20P2 and CF340G clustered with three *A. phagocytophilum* reference sequences (GQ412337.1, HG916767.1, AB196721.1, and U02521.1) in the *16S rRNA* gene phylogeny (Figure 5a) and with reference sequences CP035303 and CP015376.1 in the *groEL* gene phylogeny (Figure 5b). Sample sequence CF519 was grouped in the same cluster as the *Ca*. Anaplasma boleense reference sequence (KX987334.1) for the *16S rRNA* gene phylogeny (Figure 5a) and CP035303 and CP015376.1 for the *groEL* gene (Figure 5b). Phylogeny based on *Wolbachia* spp. maximum likelihood analysis for the *16S rRNA* gene indicated that the sequence of sample CF472 clustered in the same group with the *Wolbachia* spp. endosymbiont reference sequence (KP114101.1) (Figure 5a).

A phylogenetic analysis of sequences based on *16S rRNA* genes revealed that samples CF393, DN387, and DN387G, as well as AM1A, clustered in the same group with reference sequences from *E. canis* (NR.118741.1, MG967466.1, and EF195135.1) (Figure 5a). The *groEL* gel phylogeny demonstrated that the sequences of samples CF393, DN387, and DN387G clustered in the same group with the *E. canis* reference sequence (OP000107.1 and OP006708.1) (Figure 5b).

## 4. Discussion

Based on the limited knowledge about TBDs in Paraguay [43,44], an epidemiological-molecular study was conducted on bacteria from the Anaplasmataceae family. This study involved the analysis of tick, domestic dog, and wild armadillo specimens collected from different forest areas and indigenous and rural communities in the north of the country. Blood meal sources of free-living ticks were also investigated.

The ticks identified in this study belong to the Ixodidae family, specifically the genera *Rhipicephalus* spp. and *Amblyomma* spp. Previous reports in the country have described the presence of the same tick species found in this study on different hosts, as follows: *R. sanguineus* in urban dogs [45]; *A. ovale* in dogs, humans, birds, and free-living ticks [23]; *A. coelebs* in dogs and humans [23,45]; and *A. sculptum* parasitizing various wildlife species, such as deer, peccaries, cougars, panthers, and accidentally, humans [23,45]. This study reports the parasitism of wild armadillos with *A. sculptum* and *A. coelebs* for the first time in the country.

Despite the existence of several studies on the food sources of ticks collected from both domestic and wild animals [46,47,48,49], reports on free-living ticks remain scarce [50]. To better understand the vector-host dynamics and due to its epidemiological importance, the blood meal sources of free-living ticks were investigated by detecting vertebrate DNA in their intestinal contents. For this purpose, the *cyt b* gene was selected as the target due to its high copy number. It effectively identified the blood meal source of hematophagous arthropods, including ticks [46,47,48,49,50]. To reduce contamination, particularly human DNA, we sterilized the surface of each sample with sodium hypochlorite, as previous studies have demonstrated the effectiveness of this protocol [34,51]. This study is the first in the country in which free-living ticks of the genus *Amblyomma*, hematophagous ectoparasites that feed on mammal, amphibian, reptile, and bird blood were collected [52]. Our molecular results showed that free-living ticks of the genus *Amblyomma* fed most frequently on rodent blood, followed by human, armadillo, and dog blood. This study demonstrates dynamic foraging patterns, highlighting the presence of human and canine blood, and provides valuable preliminary information on the host diversity on which free-living *Amblyomma* spp. ticks feed.

The traditional methods used to approach TBD studies have varied with the advances, development, and availability of new techniques, including culture, serologic methods, various sequencing methods, such as NGS and pyrosequencing, and PCR techniques and their variants [53,54]. One such variant is the application of the method of high-resolution melting (HRM) analysis, which is an effective technique used for the detection and differentiation of VBDs [53,55]. This technique has been implemented in a variety of pathogens and with various approaches, including the determination of discrete typing units (DTUs) of *Trypanosoma cruzi* [56], the genotypic differentiation of *Leishmania* spp. parasites [57], bacterial identification [58], fungal identification [59], and more recently, the differentiation of resistant variants of *Aedes aegypti* [60]. In this study, a real-time PCR with HRM analysis was optimized to investigate bacteria of the Anaplasmataceae family. Different melting values were obtained for *Ehrlichia* spp. and *Anaplasma* spp. to clearly differentiate between them. Amplified products were sequenced, and the identified species were found to include *E. canis*, *A. platys*, *A. phagocytophilum*, *Ca*. A. boleense, and *Wolbachia* spp.

*E. canis* is the causative agent of canine monocytic ehrlichiosis. This multisystemic disease can present with subclinical, acute, or chronic manifestations, depending on the pathogenicity of the strain, co-infections with other pathogens, or host predisposition [61]. *R. sanguineus* sensu lato (s.l.) is the main vector for this pathogen [62]. The prevalence of *E. canis* in dog blood in this study is similar to that reported in Argentina, Brazil, and Paraguay [24,43], except for a more recent study in Paraguay that found higher prevalences in dogs from urban areas [25]. *E. canis* has a worldwide distribution and is considered an endemic pathogen in almost every continent [63]. For many decades, Australia was thought to be the only region of the world where this pathogen was absent. However, the scenario has changed since 2020, when the first detection of *E. canis* in several dogs in Western Australia was reported [64]. In addition, this pathogen has been reported with wide distribution in wild hosts, such as canids, felids, birds, and free-living ticks [65]. The results of this study provide the first molecular evidence in the country for the presence of *E. canis* in the blood of wild armadillos, *R. sanguineus* collected from dogs, and *A. sculptum* collected from armadillos and vegetation. *E. canis* has also been identified in ticks collected from foxes and coatis from Brazilian Pantanal [65]. In addition, there is molecular evidence of human infections in Venezuela and Costa Rica, positioning it as a potential zoonotic agent [66,67,68].

*A. platys* is the causative agent of canine infectious cyclic thrombocytopenia, and *R. sanguineus* s.l. is the confirmed vector of this pathogen [69]. The prevalence of *A. platys* in dog blood in the present study was lower than that reported in dogs from the urban areas of Argentina, Brazil, and Paraguay [25,43,70]. We reported, for the first time, the presence of *A. platys* in armadillos, *R. sanguineus* collected from dogs, and strikingly, many free-living *A. sculptum* specimens.

*A. phagocytophilum*, the causative agent of human granulocytic anaplasmosis, is regarded as an emerging pathogen with significant implications for public and veterinary health [71]. This species has complex zoonotic cycles involving wild hosts and various tick species [72] and has even been detected in *A. sculptum* free-living ticks [65]. In this study, *A. phagocytophilum* was detected, for the first time in the country, in the blood of domestic dogs, as well as in *R. sanguineus* collected from the same animals, although at a lower prevalence than that reported in a study of domestic animals in Brazil [43,73]. Positive samples of this pathogen have been collected in dogs from Aché and Ava Guaraní indigenous communities, which still maintain part of their traditional ancestral activities, such as hunting, during which they are accompanied by their dogs [74]. Further research is necessary to determine the potential public health implications of *A. phagocytophilum* in dogs and their ticks in various parts of the world.

*Candidatus* Anaplasma boleense was first detected in *Hyalommma asiaticum* and *Rhipicephalus microplus* from China [75]. Subsequently, findings of this bacterium were reported in various parts of the world, in different arthropods and ruminants, including mosquitoes of China [76], cattle and deer from Malaysia [77], and cattle from Ethiopia [78], Mozambique [79], and South Africa [80]. In Latin America, recent studies have also detected *Ca*. A. boleense in wild deer [81], goats, *R. microplus* [82], and *Amblyomma tigrinum* [83] from Argentina and goats [84] from Brazil. The present study has identified the first occurrence of *Ca*. Anaplasma boleense in Paraguay domestic dog population, both in the blood and *R. sanguineus* ticks.

This may be due to the interaction of these free-living canids with the ruminants raised by Indigenous communities without veterinary care. Given the lack of knowledge regarding the pathogenicity and zoonotic importance of this bacterium, the detection of the bacterium in ticks parasitizing domestic dogs, such as *A. tigrinum* [83] and *R. sanguineus*, as reported in this study, highlights the necessity for further epidemiological studies employing a “One Health” approach.

*Wolbachia* spp. are endosymbiotic bacteria that belong to the ancestral clade of *Anaplasma* spp. and *Ehrlichia* spp. [85] and coexist in mutualistic symbiosis with filariae (nematodes), which act as hosts for the development of this bacterium [86]. Recent studies have demonstrated that detecting both *Wolbachia* spp. and filarial DNA simultaneously improves the diagnosis of these infections [86]. During our study, we incidentally identified *Wolbachia* spp. DNA in the bloodstream of dogs, marking the first detection of *Wolbachia* spp. in the country. Positive samples were found in all but one of the sites sampled, indicating a wide distribution in the study area. Further studies are necessary to gain a better understanding of filariasis, which is a severe disease that affects dogs and can also be transmitted to humans.

The results of the phylogenetic analysis for all bacteria identified in this study indicate a clear separation of well-defined lineages for the following species: *E. canis*, *A. platys*, *A. phagocytophilum*, *Ca*. A. boleense, *Anaplasma* spp., and *Wolbachia* spp. Some isolates show differences within the groups compared to the reference sequences used, indicating the genetic diversity of Anaplasmataceae bacteria circulating in hosts and ticks in the study area. However, further assessment using additional genetic markers is necessary.

After determining the bacterial species through sequencing, we analyzed the denaturation curves. Each curve produced different values, indicating the potential use of the optimized PCR-HRM technique for identifying Anaplasmataceae family species. A previous study employed multiplex PCR-HRM and the *16S rRNA* marker to identify and differentiate pathogenic species, including *E. canis* and *A. platys* [53]. The results demonstrated consistent T_m_ values comparable to those observed in our previous experiments [53]. Nevertheless, further analyses are necessary to evaluate the potential overlap of the melting curves with other bacterial species belonging to the Anaplasmataceae family, such as *A. marginale*, *A. centrale*, *A ovis, A. bovis*, *E. chaffeensis*, *E. ewingii*, *E. ruminantium*, *E. muris*, and *E. minasensis*.

## 5. Conclusions

This study has identified a high prevalence of Anaplasmataceae bacteria, including zoonotic species, among domestic dogs, armadillos, and ticks in the area. The molecular detection of *A. phagocytophilum*, *Ca*. A. boleense, and *Wolbachia* spp. is reported for the first time in Paraguay. Wild armadillos and free-living *A. sculptum* ticks were found to be carrying *E. canis* and *Anaplasma* spp. All tested dog populations were found to be infected with at least one pathogen, confirming their role in maintaining and transmitting ectoparasites and associated pathogens, bridging natural and anthropogenic environments. The high prevalence of *Wolbachia* spp. in dogs shows the need for further investigation into associated nematode parasites. Additional research is required to understand the epidemiological implications of tick-borne pathogens in the MFBR and neighboring communities in northeastern Paraguay.

## Figures and Tables

**Figure 1 microorganisms-12-01893-f001:**
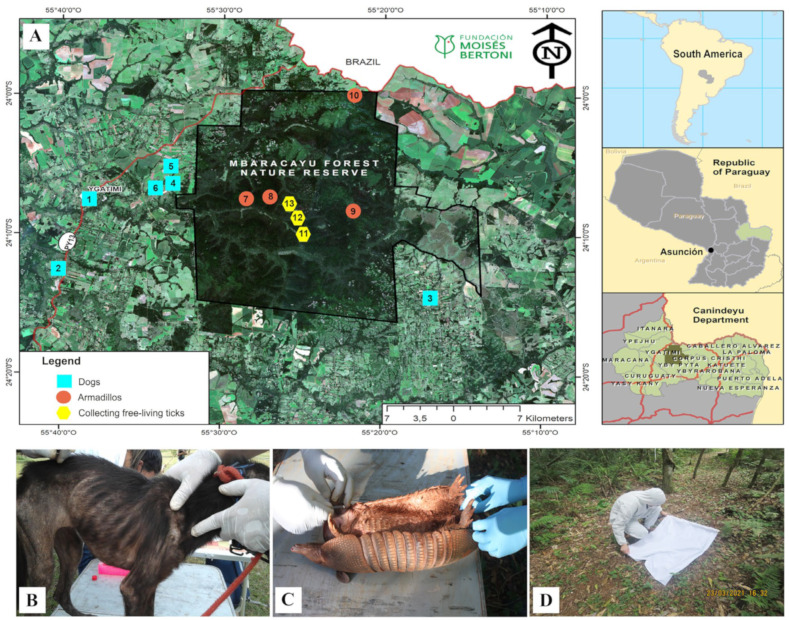
Study area and sample collection. (**A**) Sampling sites in the RBBM in Northeastern Paraguay included domestic dogs (sites 1–6), wild armadillos (sites 7–10), and free-living ticks (sites 11–13); (**B**,**C**) show a representative domestic and wild host, respectively; and (**D**) illustrates the method for collecting free-living ticks.

**Figure 2 microorganisms-12-01893-f002:**
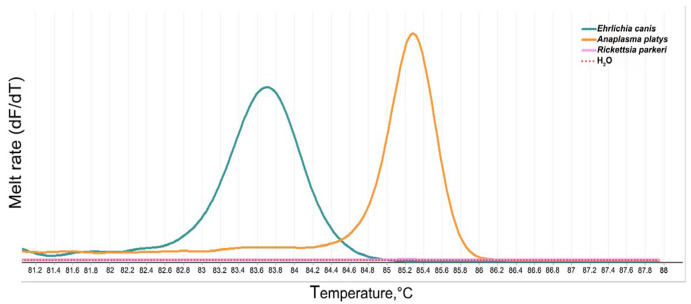
PCR amplicon melting rates for *E. canis* and *A. platys* DNA were used as positive controls.

**Figure 3 microorganisms-12-01893-f003:**
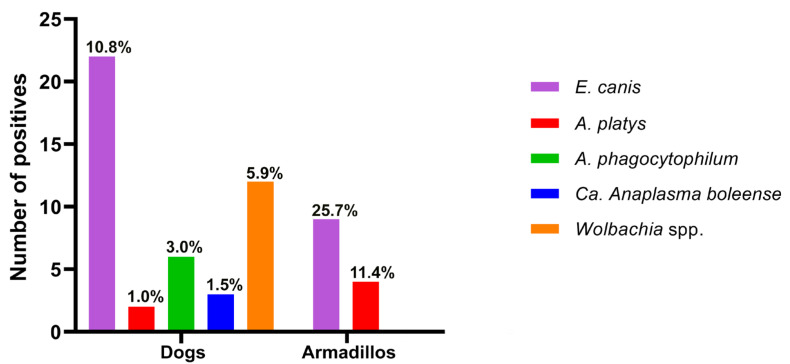
Bacteria species from the Anaplasmataceae family were detected in the blood of domestic and wild hosts.

**Figure 4 microorganisms-12-01893-f004:**
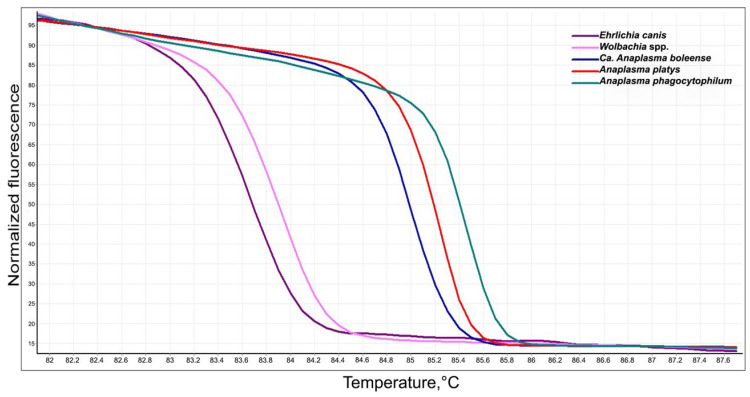
Normalized HRM curve of Anaplasmataceae bacterial species diversity detected in Paraguay.

**Figure 5 microorganisms-12-01893-f005:**
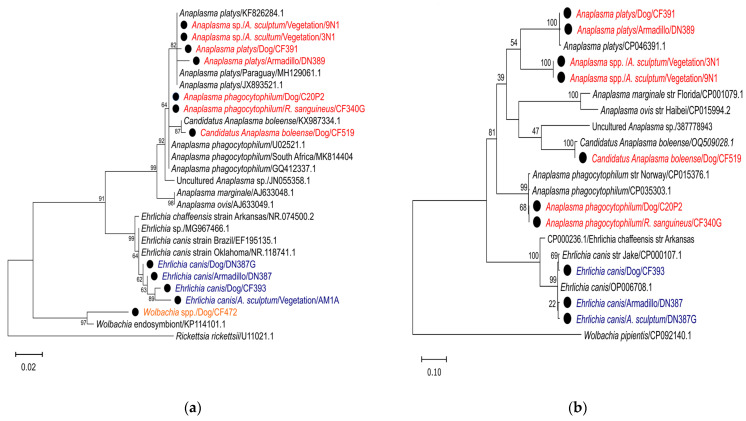
Maximum likelihood phylogenetic trees reconstructed based on the (**a**) *16S rRNA* and (**b**) *groEL* gene sequences of Anaplasmataceae, including those obtained in this study and other known sequences. The black dot represents the sequences determined in this study.

**Table 2 microorganisms-12-01893-t002:** Taxonomic classification of host- and free-living ticks collected in northeastern Paraguay.

Type of Habitat (*n*)	Hosts *n* (%)	Ticks (*n*)	Life Cycle				GenBank Accession Number
			F	M	N	L	
Dog (203)	23 (11.3)	*R. sanguineus* (24)	10	6	8	-	OR976129, OR976130 (1F, 1N)
		*A. ovale* (2)	2	-	-	-	OR976131 (1F)
Armadillo (35)	13 (37.1)	*A. sculptum* (39)	21	-	18	-	OR976133, OR976134 (1F, 1N)
		*A. coelebs* (4)	-	-	4	-	OR976132 (1N)
Free-living (1855)		*A. sculptum* (808)	46	24	738	-	OR976135, OR976136, OR976137 (1F, 2N)
		*Amblyomma* spp. (1047)	0	0	0	1047	
		*n*: 1924	79	30	768	1047	

F, female; M, male; N, nymph; L, larvae.

**Table 3 microorganisms-12-01893-t003:** Molecular identification of the blood meal sources in free-living ticks.

Ticks (*n*)	Life Cycle (*n*)	Total Detected	Blood Meal Source			
			Rodent	Human	Armadillo	Dog
*A.sculptum* (57)	Female (32)	12	10	-	1	1
	Nymph ^a^ (25)	10	-	5	3	2
*Amblyomma* spp. (7)	Larvae ^b^ (7)	-	-	-	-	-
n: 64	Total (*n*/%)	22 (34.4)	10 (45.5)	5 (22.7)	4 (18.2)	3 (13.6)

^a^ pool of 20 nymphs. ^b^ pool of 100 larvae.

**Table 4 microorganisms-12-01893-t004:** Prevalence of Anaplasmataceae in ticks collected from different hosts/habitats.

Host/Habitat (*n*)	Ticks (*n*)	Life Cycle (*n*)	No. of Ticks Infected with Bacteria Anaplasmataceae					Infected/ Collected (%)
			*Ehrlichia* *canis*	*Anaplasma* *platys*	*Anaplasma* *phago*.	*Ca.* Anaplasma boleense	*Anaplasma* spp.	
Dogs (26)	*R. sanguineus* (24)	Female (10)	-	2	2	1	-	9/26 (34.6)
		Male (6)	4	-	-	-	-	
		Nymph (8)	-	-	-	-	-	
	*A. ovale* (2)	Female (2)	-	-	-	-	-	
Armadillo (43)	*A.coelebs* (4)	Nymph (4)	-	-	-	-	-	10/43 (23.3)
	*A.sculptum* (39)	Female (21)	6	-	-	-	-	
		Nymph (18)	4	-	-	-	-	
Free-living (70)	*A.sculptum* (70)	Female (46)	5	-	-	-	8	15/70 (21.4)
		Male (24)	-	-	-	-	2	
	*n*: 139	Total (*n*/%)	19 (13.7)	2 (1.4)	2 (1.4)	1 (0.7)	10 (7.2)	34 (24.5)

**Table 5 microorganisms-12-01893-t005:** Percentage of minimum infection rate of Anaplasmataceae in juvenile ticks collected from vegetation.

Free-Living Ticks (Life Cycle Stage)	No. of Pools	Anaplasmataceae Positive (Minimum Infection Rate %)				
		*Ehrlichia* *canis*	*Anaplasma* *platys*	*Anaplasma phago*	*Ca.* Anaplasma boleense	*Anaplasma* spp.
*A.sculptum* (nymphs ^a^)	4	1 (1.25)	-	-	-	-
	10	-	-	-	-	2 (1.0)
	21	-	-	-	-	-
*Amblyomma* spp. (larvae ^b^)	2	-	-	-	-	-
	7	-	-	-	-	-

^a^ pool of 20 nymphs. ^b^ pool of 100 larvae.

**Table 6 microorganisms-12-01893-t006:** Identification of Anaplasmataceae species in dogs, armadillos, and ticks by sequencing the *16S rRNA* and *groEL* gene.

Sample Code	Sources	Gene	GenBank Accession Number	Species	Closest Match Accession Number, Similarity (%)
CF393	*Canis familiaris*	*16S rRNA*	OR976118	*Ehrlichia canis*	MN922610.1, (99.13)
		*groEL*	PP928796		OP006713.1, (99.72)
DN387	Armadillo	*16S rRNA*	OR976119	*Ehrlichia canis*	MN922610.1, (99.71)
		*groEL*	PP928797		OP006708.1, (99.15)
DN387G	*A. sculptum* ^a^	*16S rRNA*	OR976120	*Ehrlichia canis*	MN922610.1, (100.0)
		*groEL*	PP928798		OP006708.1, (99.15)
CF391	*Canis familiaris*	*16S rRNA*	OR976123	*Anaplasma platys*	MN630836.1, (99.13)
		*groEL*	PP928804		CP046391.1, (99.43)
DN389	Armadillo	*16S rRNA*	OR976122	*Anaplasma platys*	MN630836.1, (99.10)
		*groEL*	PP928805		CP046391.1, (99.14)
3N1	*A. sculptum*/nymph ^b^	*16S rRNA*	PP930511	*Anaplasma* spp.	MT019534.1, (100.0)
		*groEL*	PP928799		LC381241.1, (85.01)
9N1	*A. sculptum*/nymph ^b^	*16S rRNA*	PP930512	*Anaplasma* spp.	MT019534.1, (100.0)
		*groEL*	PP928799		LC381241.1, (85.01)
C20P2	*Canis familiaris*	*16S rRNA*	OR976125	*Anaplasma phagocytophilum*	CP015376.1, (99.42)
		*groEL*	PP928801		OQ453812.1, (100.0)
C20P2G	*R. sanguineus* ^c^	*16S rRNA*	OR976126	*Anaplasma phagocytophilum*	CP015376.1, (99.42)
		*groEL*	PP928802		OQ453812.1, (100.0)
CF519	*Canis familiaris*	*16S rRNA*	OR976128	*Ca*. A. boleense	KX987335.1, (99.71)
		*groEL*	PP928803		OQ509028.1, (99.14)
CF472	*Canis familiaris*	*16S rRNA*	OR976117	*Wolbachia* spp.	MT792375.1, (99.13)

^a^ free-living. ^b^ collected from armadillo. ^c^ collected from dog.

## Data Availability

All data generated or analyzed during this study are included in this published article.

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
