# Peer review of "Diversity of Anaplasmataceae Transmitted by Ticks (Ixodidae) and the First Molecular Evidence of Anaplasma phagocytophilum and Candidatus Anaplasma boleense in Paraguay"

_microorganisms, 2024, doi:10.3390/microorganisms12091893_

Round 1
Reviewer 1 Report
Comments and Suggestions for Authors
In this study, the authors undertake surveillance for Anaplasmataceae bacteria, a family that contains many important tick-borne pathogens, in a region of Paraguay, a country in which few tick-borne disease studies have taken place. This research collected ticks and blood samples from dogs and armadillos, sampled free-living ticks, and tested these for presence of Anaplasmataceae using standard PCR detection methods. The authors also developed a high resolution melting curve method to separate genus Anaplasma and Ehrlichia, as well as different species within these genera. Sequencing and phylogenetic analysis of 16S rRNA and groEL markers was also performed to identify the different bacterial species detected. Morphological identification of ticks was supplemented with molecular confirmation and bloodmeal analysis was performed to identify hosts on which free-living ticks fed.
Overall this is a thoroughly conducted study and advances knowledge of the ticks, their hosts, and presence of Anaplasmataceae in this region of Paraguay. The study improves knowledge of circulating bacteria, some of which are confirmed or potential threats to human and animal health, in a relatively understudied region of the world. This paper will be of interest to tick-borne disease and One Health researchers, particularly those in Latin America.
I only have minor concerns about this manuscript, mainly relating to clarification of a few points, detailed below:
1. line 41 - correct to triatomine bugs
2. line 101-102: since the armadillos and dogs were not collected, it would be more accurate to use "sampled" rather than "collected" in this sentence.
3. line 113: 15-20 mins is a long time in between tick checks of the flag - it is usual to check for ticks every 5-10 metres of dragging.
4. line 124/Table 2: It is surprising that larvae were not identified molecularly since they are hardest to identify morphologically.
5. line 193: what is the Genbank number for A. ovis 16S rRNA?
6. line 209: correct to Microsoft Office
7. line 241: clarify whether the 116 samples were pooled samples. Should it say "116 pooled samples"?
8. line 288: put 5 females in brackets ()
9. Table 4: it is not correct to call animals habitats, so perhaps "different habitats" should be corrected to "different hosts/habitats", and "habitat type" should be changed to "host/habitat" or "tick collection type"
10. Table 5: change "life cycle" to "life cycle stage".
It is unclear in the table what the numbers 1 / 1.25 refer to. Instead of Anaplasmataceae Minimum Infection Rate (%), the heading could say Anaplasmataceae positive (Mininimum infection rate %), and then give numbers as 1 (1.25), 2 (1.0)
11. line 343: remove brackets from (DN389), (CF391), (9N1) and (3N1).
12. line 375-6: change all the "in" to "on". Ticks are found on animals, not in them.
13. line 420: correct to ".. and A. sculptum collected from armadillos and vegetation."
14. line 421: in foxes and coatis from which countries/regions of the world?
Comments on the Quality of English LanguageLanguage is overall good and easy to read and understand. There are some areas where grammar/spelling can be improved:
1. line 32: delete "blood" from end of sentence.
2. line 51: infectious agents
3. line 87: delete "Tintel et al"
4. Table 3: correct spelling of Blood in table.
5. line 381-3: grammar needs to be improved in this sentence. Maybe it is just missing "we investigated" or can be changed to "...the blood meal sources of free-living ticks were investigated by detecting vertebrate DNA in..."
6: line 445: change "such as" to "and"
7. line 449: change "within the country" to "in Paraguay"
8. line 485: change "in the country" to "in Paraguay"
9: line 486-488: grammar of this sentence is not quite right. Should it say "...confirming their role in maintaining and transmitting ectoparasites and associated pathogens, ..."
Author Response
A response letter has been added to identify each reviewer's requests, with a detailed list of corrections

Reviewer 2 Report
Comments and Suggestions for Authors
The manuscript presents a study where the objective was to investigate the occurrence and diversity of Anaplasmataceae bacteria in ticks, domestic and wild hosts, as well as the tick diversity in different habitats in the Mbaracayú Forest Biosphere Reserve (MFBR) reserve in Paraguay. The study also determines the identification of blood meal sources of free-living ticks using molecular techniques.
While there is no novelty in this study in terms of the experimental tools used for estimating the molecular epidemiological prevalence, the study is of interest for regional analysis as it is shown that the DNA of tick-borne pathogens (Ehrlichia canis, Anaplasma platys, Anaplasma phagocytophilum and Candidatus Anaplasma boleense) was identified in the blood of sampled domestic dogs. The study also describes the finding of pathogens DNA in Rhipicephalus ticks associated with these hosts and in free-living Amblyomma ticks collected in the northeast region of Paraguay. This result makes the authors claim the first molecular evidence of the pathogens E. canis and A. platys in armadillos in this area. The results contribute to the growing understanding of the role of wildlife as pathogen reservoirs and carriers of pathogen-infected ticks, which may offer valuable insights into the epidemiology of vector-borne diseases, providing a foundation for reducing the risk of tick-borne diseases.
Overall, the manuscript is well written, the laboratory approach concerning the analytical diagnostic procedures utilized is adequate, and sufficient technical details are provided to replicate the work. The results are presented clearly and, in enough detail, to support the conclusions.
Minor details
Line 23. Spell PCR-HMR, first-time use in the manuscript. Use a comma instead of a period in “…optimized for this purpose. while the…”
Line 26. Add a comma in between “…included Ehrlichia canis Anaplasma platys,”
Line 28. Add “(Canis lupus familiaris)” after “domestic dogs”
Line 132. Correct “…by treating it with it with 1%...”
Line 146. Check legend to Table. Table 2 should be Table 1.
Line 181. Add “with” in-between “…purified GeneJET…”
Line 232. Check “(Table 1)”, it should be Table 2
Line 363. Check legend to Figure. Figure 2 should be Figure 5.
Line 401. Spell “DTUs”, first-time use
Line 416. It is stated that “E. canis…is considered an endemic pathogen in almost every continent except Australia [62]”. Up until recently, as the scenario has changed, see for example: Matthew J. Neave, Patrick Mileto, Ancy Joseph, Tristan J. Reid, Angela Scott, David T. Williams, Anthony L. Keyburn. Comparative genomic analysis of the first Ehrlichia canis detections in Australia. Ticks and Tick-borne Diseases. Volume 13, Issue 3, 2022, 101909, https://doi.org/10.1016/j.ttbdis.2022.101909.
Line 420. Correct the sentence “…collected from dogs, A. sculptum. collected from armadillos, and free-living.”
Lines 430-431. Delete the period in “…in dogs and their ticks. from various parts…”
Line 480. Add “of” in “…bacterial species the Anaplasmataceae family”. Please, indicate what other species (Ehrlichia chaffensis, Ehrlichia ewingii….?).
Line 487. Revise the sentence, something is missing in “for at least one pathogen, their role in maintaining and transmitting ectoparasites and associated pathogens…”
Line 519. Use “accessed on Day Month Year” after the URL, instead of [cited 2024 April 30]
Line 521. The book title is repeated. Add Chapter title (Chapter 4 Vector-Borne Diseases).
See instructions for authors. example: Title of the chapter. In Book Title, 2nd ed.; Editor 1, A., Editor 2, B., Eds.; Publisher: Publisher Location, Country, Year; Volume 3, pp. 154–196.
Line 524. Check for correct citation. Journal Articles: 1. Author 1, A.B.; Author 2, C.D. Title of the article. Abbreviated Journal Name Year, Volume, page range.
Lines 693-694. Delete Journal title “Comparative Immunology, Microbiology and Infectious Diseases“
Comments on the Quality of English LanguageMinor editing of the English language is required
Author Response

(The authors gave the same response as above.)

Reviewer 3 Report
Comments and Suggestions for Authors
Dear Authors,
I reviewed your manuscript "Diversity of Anaplasmataceae Transmitted by Ticks (Ixodidae) and First Molecular Evidence of Anaplasma phagocytophilum and Candidatus Anaplasma boleense in Paraguay". It was a tough job for me! See my comments below:
Abstract section
Line 23: There is a dot: "purpose. while" Should be a comma! This is a typo!
Line 26: „canis, Anaplasma”
Lines 27-29: reworded: "Free-living and parasitic ticks on dogs and wild armadillos (Dasypus novemcinctus) were collected and identified as Rhipicephalus sanguineus and Amblyomma spp."
Introduction section
Lines 48-51: rewritten in active voice: "Regarding ticks, in addition to the epidermal damage that toxins in their saliva can cause, they are considered one of the most relevant arthropods in public health, surpassed only by mosquitoes in their role as human pathogen vectors due to their ability to transmit infectious agent [5]."
Lines 52-55: I split your extended paragraph into two more intelligible sentences, as follows: "They are classified as arthropods of the class Arachnida, divided into two main families: Argasidae (soft ticks) and Ixodidae (hard ticks). The world's largest number of species reported belong to the last family, including the most medically important species [5,6]."
Lines 59-62: I reorganized your paragraph: "The vast repertoire of pathogens transmitted by ticks includes bacteria. In this context, it is essential to emphasize that these arthropods play a fundamental role in the evolution and transmission of the Anaplasmataceae family bacteria, causative agents of emerging infectious diseases [8]."
Lines 70: „features”
Lines 75-76: „such as PCR, serological methods, and eco-epidemiological studies [10,20–22].”
Lines 90-92: reorganized: "Based on the gap in knowledge regarding TBDs, we conducted this research in the Mbaracayú Forest Biosphere Reserve (MFBR). This protected area represents two ecosystems inhabited by indigenous Aché and Guaranícommunities [27]."
Line 95: "This study aimed to..." instead of "The objective of this study was to..."
Materials and Methods
Line 101: use the plural form: "...domestic dogs (Canis lupus familiaris), and wild armadillos..."
Lines 101-108: This paragraph includes three sentences that mention almost identically the same things. Please seriously rephrase this paragraph, avoiding repetitions!
Lines 108-113: Your description is not very clear! First, there is a slight difference between flagging and dragging; did you perform dragging? Second, you walked for 15 minutes (a line of 1 km distance) in one direction, stopped, collected the ticks, and then turned back in the opposite direction, repeating? If yes, I don't understand this statement: "Ticks were collected for 50 minutes." Would you rephrase your description in a clearer and more intelligible way?
Line 120: use „Identification” instead of „Classification”
Lines 120-122: reworded: "Adult ticks were identified using a dichotomous key [29,30], whereas the nymphs and larvae of the Rhipicephalus and Amblyomma genera, based on specific keys [31,32]."
Line 122: use „identification”, not „classification”. Ticks are already classified!
Lines 124-125: reworded: "We selected and amplified at least two specimens for each of the identified genera's life stages (adult and nymph)."
Lines 134-137: reworded: "All ticks collected from hosts were processed individually. Free-living ticks were processed differently: the adults individually, nymphs in pools of 20, and larvae in pools of about 100 individuals."
Results section
Lines 227-237: You must rewrite this paragraph to be easier to understand, avoid repetitions, and be organized!
Just an example:
"A total of 1924 ticks (79 females, 30 males, 768 nymphs, and 1047 larvae) were collected and identified, 1855 of which were free-living ticks, and 69 were host-collected. Out of 203 examined dogs, 23 (11.3%) were tick-infested; 26 specimens were collected, 24 identified as Rhipicephalus sanguineus (10 females, six males, and eight nymphs), and two females as Amblyomma ovale."...and so on!
A general comment related to this section: You describe the results obtained in a very convoluted, twisted, and complicated style. Short, concise phrases without repetition are desirable!
Discussion section
Lines 372-373: Don't repeat "dogs, armadillos, and free-living ticks"! You already mentioned this a few lines above! Actually, your manuscript has plenty of repetitions!
Lines 378-379: This is a better option: "This study reports the parasitism of wild armadillos with A. sculptum and A. coelebs for the first time in the country." since both species were already registered in Brazil on armadillos.
Line 380: I disagree with your statement, "There are limited reports on the food sources of free-living ticks around the world." In five minutes, I found six published articles (see at the end of my comments); you referenced none!
Lines 381-383: this sentence has no meaning: "To better understand the vector-host dynamics and, due to its epidemiological importance, investigated the blood meal sources of free-living ticks by detecting vertebrate DNA in their intestinal contents." Who "investigated"?
Lines 383-385: reworded and reorganized sentence: "For this purpose, the cyt b gene was selected as the target due to its high copy number. It effectively identified the blood meal source of hematophagous arthropods, including ticks."
Lines 409-411: reworded and reorganized: "E. canis is the causative agent of canine monocytic ehrlichiosis. This multisystemic disease can present subclinical, acute, or chronic manifestations depending on the pathogenicity of the strain, co-infections with other pathogens, or host predisposition [60]."
Lines 412-415: Write the prevalence values in this sentence to better see the comparison: "The prevalence of E. canis in dog blood in this study is similar to that reported in Argentina, Brazil, and Paraguay [24,43], except for a more recent study in Paraguay that found higher prevalences in dogs from urban areas [25]."
Line 420: "...and free-living ticks."!
Line 424: use italics for „A. platys”!
Lines 425-427: Write the prevalence values here: "The prevalence of A. platys in dog blood in the present study was lower than that reported in dogs from urban areas of Argentina, Brazil, and Paraguay [25,43,68]." to really see this comparison!
Line 441: cut the dot here: "ticks. from"
Lines 463: use italics for „Wolbachia”
Conclusion section
Line 485: cut the dot here: „sculptum. ticks”
Allan BF, Goessling LS, Storch GA, Thach RE. Blood meal analysis to identify reservoir hosts for Amblyomma americanum ticks. Emerg Infect Dis. 2010 Mar;16(3):433-40. doi: 10.3201/eid1603.090911.
Che Lah EF, Ahamad M, Haron MS, Ming HT. Blood meal identification of field collected on-host ticks surrounding two human settlements in Malaysia. The Experiment. 2013; 17(2): 1177-1183
Che Lah EF, Yaakop S, Ahamad M, Md Nor S. Molecular identification of blood meal sources of ticks (Acari, Ixodidae) using cytochrome b gene as a genetic marker. Zookeys. 2015 Jan 28;(478):27-43. doi: 10.3897/zookeys.478.8037.
Kim HJ, Hamer GL, Hamer SA, Lopez JE, Teel PD. Identification of Host Bloodmeal Source in Ornithodoros turicata Dugès (Ixodida: Argasidae) Using DNA-Based and Stable Isotope-Based Techniques. Front Vet Sci. 2021 Feb 17;8:620441. doi: 10.3389/fvets.2021.620441.
Önder Ö, Shao W, Kemps BD, Lam H, Brisson D. Identifying sources of tick blood meals using unidentified tandem mass spectral libraries. Nat Commun. 2013;4:1746. doi: 10.1038/ncomms2730.
Oundo JW, Villinger J, Jeneby M, Ong'amo G, Otiende MY, Makhulu EE, Musa AA, Ouso DO, Wambua L. Pathogens, endosymbionts, and blood-meal sources of host-seeking ticks in the fast-changing Maasai Mara wildlife ecosystem. PLoS One. 2020 Aug 31;15(8):e0228366. doi: 10.1371/journal.pone.0228366.
Comments on the Quality of English LanguageEnglish also needs a significant revision regarding commas, dots, verb tenses, lack of articles, clarity, and so on!
Author Response

(The authors gave the same response as above.)
